# Research on using *Aquilaria sinensis* callus to evaluate the agarwood-inducing potential of fungi

Zhikai Wang[1,2,3], Guoying Zhou[1,2,3], Jungang Chen[1,2,3], Xinyu Miao[1,2,3], Yandong Xia[1,2,3], Zhuang Du[1,2,3], Junang Liu[1,2,3]*

1 Key Laboratory of National Forestry and Grassland Administration for Control of Diseases and Pests of South Plantation, Central South University of Forestry and Technology, Changsha, China, 2 Key Laboratory of Cultivation and Protection for Non-Wood Forest Trees, Central South University of Forestry and Technology, Changsha, China, 3 Hunan Provincial Key Laboratory for Control of Forest Diseases and Pests, Central South University of Forestry and Technology, Changsha, China

* kjc9620@163.com

**Data Availability Statement:** All relevant data are within the manuscript and its Supporting Information files.

**Funding:** This research was funded by National Key R & D Program of China (2023YFD1401304).

## Abstract

Agarwood is a precious resinous heartwood highly valued for its cultural, religious, and medicinal significance. With the increasing market demand, natural agarwood resources are rapidly depleting, making the development of effective artificial induction methods for agarwood highly significant. This study aims to explore the feasibility of using callus tissue to assess the ability of fungi to induce agarwood formation. We selected two fungi isolated from *Aquilaria sinensis*, W-1 (*Podospora setosa*) and W-15 (*Alternaria alstroemeriae*), and used the known agarwood-inducing fungi YMY (*Pestalotiopsis* sp.) as a positive control, by treating *A. sinensis* callus with their fermented filtrates. The experimental results showed that W-1 and W-15 treatments significantly enhanced the activity of Superoxide dismutase (SOD) and Peroxidase (POD) in the callus tissue and upregulated the expression of 3-hydroxy-3-methylglutaryl-CoA synthase (*HMGS*), 1-deoxy-D-xylulose-5-phosphate reductoisomerase (*DXR*), and sesquiterpene synthase (*ASS-1*). GC-MS analysis further confirmed that the contents of sesquiterpenes and aromatic compounds in *A. sinensis* treated with W-1 and W-15 were significantly elevated, suggesting that these fungi possess the capacity to induce the formation of agarwood. This study demonstrates that using callus tissue to screen fungi capable of inducing agarwood is feasible and effective, providing new insights for screening fungi resources that efficiently induce agarwood formation in the future.

## Introduction

Agarwood is a highly valuable dark resinous heartwood formed in *Aquilaria* species following lightning strikes, animal gnawing, or pest attacks [1]. Due to its cultural, religious, and medicinal significance [2,3], the demand for agarwood has continually increased, leading to over-exploitation and illegal trade of *Aquilaria* trees, which resulted in its listing under Appendix II

'The funders had no role in study design, data collection and analysis, decision to publish, or preparation of the manuscript.

of the Convention on International Trade in Endangered Species of Wild Fauna and Flora (CITES) in 1995 [4]. To prevent the extinction of *Aquilaria sinensis* and to meet market demand, methods such as physical injury, chemical induction, and microbial inoculation have been used to artificially induce agarwood formation in *A. sinensis*. Compared to physical and chemical induction methods, microbial agents treeare considered to be capable of intermittently and continuously inducing agarwood formation [5]. Many scientists have been dedicated to isolating various fungi from different parts of *Aquilaria* trees, and have confirmed that *Trichoderma* [6], *Aspergillus niger* [7], *Fusarium*, *Lasiodiploidia* [8] are capable of inducing agarwood formation.

Fungi, as the key factor in inducing agarwood formation, stimulate *A. sinensis* to produce defensive secondary metabolites through intricate physiological and biochemical mechanisms. In a natural environment, when agarwood trees are infected by fungi, the tree recognizes the fungi invasion and activates its defense response, a process considered the starting point of agarwood formation [9]. The core of the defense response lies in inducing stress responses through signaling pathways, leading to enhanced antioxidant enzyme activity, triggering the expression of a series of disease-resistant genes, and ultimately resulting in the accumulation of secondary metabolites [2]. These metabolites, including sesquiterpenes and 2-(2-phenylethyl) chromones, are the main active components of agarwood [10].

Fungi infection is often accompanied by oxidative stress responses in plant cells. Plant cells counter pathogen invasion by producing reactive oxygen species (ROS) such as superoxide anions ($O_2^-$) [11]. To mitigate the toxicity of ROS, the antioxidant enzyme system within agarwood trees is rapidly activated. Superoxide dismutase (SOD), Peroxidase (POD), and Catalase (CAT) are the main antioxidant enzymes involved, working together to eliminate excess ROS and maintain redox balance within cells [12]. The enhancement of these enzyme activities is not only a direct response of plants to fungi invasion but also indirectly promotes the activation of secondary metabolic pathways related to agarwood formation [9]. Under fungi induction, the secondary metabolic pathways within agarwood trees are significantly activated, particularly the mevalonate (MVA) pathway and the methylerythritol phosphate (MEP) pathway. 3-Hydroxy-3-methylglutaryl-CoA synthase (*HMGS*) and 1-deoxy-D-xylulose-5-phosphate reductoisomerase (*DXR*) are the key rate-limiting enzymes in the MVA and MEP pathways, respectively [13,14]. The upregulated expression of these enzymes promotes the increased synthesis of sesquiterpene compounds. Sesquiterpene synthase *(ASS-1)* is a typical wound-induced gene responsible for sesquiterpene formation in agarwood. Studies have found that *ASS-1* expression is barely detectable in healthy *A. sinensis* callus or cell cultures, but its expression significantly increases in wounded tissues or jasmonic acid methyl ester-treated callus, leading to a corresponding increase in sesquiterpene compounds [15]. Fungi infection can significantly increase the accumulation of secondary metabolites in *Aquilaria* trees by regulating the gene expression of defense enzymes and these key enzymes, thereby enhancing the yield and quality of agarwood [16].

The callus of *A. sinensis* is an undifferentiated cell mass induced from plant tissue through in vitro culture, and it shares physiological and biochemical similarities with the whole tree. The metabolic pathways and stress responses of callus tissue largely retain the characteristics of the tree, making it a common model for studying the formation mechanisms of agarwood [17,18]. However, because callus tissue grows in a controlled environment, lacking the complex structure and environmental interactions found in mature trees, its research findings may require further validation and optimization when applied to actual trees [19]. Therefore, we consider whether callus tissue can be used to screen fungi that promote agarwood formation.

In this study, we selected the fungi W-1 and W-15, whose ability to promote agarwood formation was not previously established, and used the known agarwood-inducing fungi YMY

[20] as a positive control. We treated the callus tissue and preliminarily explored the feasibility of using callus tissue to assess the ability of fungi to induce agarwood formation, providing a new approach for screening fungi that induce agarwood.

## Materials and methods

### Generation of callus

Based on the induction formula described by Liu et al [21] modifications were made to the Murashige and Skoog (MS) basal salt medium by supplementing it with 1 mg/L NAA and 2 mg/L 6-benzylaminopurine to induce callus formation in *A. sinensis*. To maintain the growth of the callus, subcultures were performed every 4 weeks using fresh callus induction medium (S1 Fig).

### Isolation and identification of fungi

Fungi were isolated from *A. sinensis* in Chengmai County, Hainan. The fungi isolates were purified by culturing on PDA plates at 28˚C for 5 days and using hyphal tip isolation technique. Genomic DNA was extracted from the fungi isolates using the Fungi Genomic DNA Extraction Kit (Beijing Solarbio Science & Technology Co., Ltd.) according to the manufacturer's instructions. The ITS region was amplified by PCR using universal primers ITS1 (5'-TCCGTAGGTGAACCTGCGG-3') and ITS4 (5'-TCCTCCGCTTATTGATATGC-3'). The PCR products were sent to Sangon Biotech (Shanghai) Co., Ltd. for sequencing. The obtained sequences were searched against the GenBank database. Phylogenetic analysis of the isolated fungi was performed using Mega 11.0 software with 1000 bootstrap replicates.

### Callus treatment

The fungi used in this experiment included W-1 (*Podospora setosa*), W-15 (*Alternaria alstroemeriae*), and YMY (*Pestalotiopsis* sp.). These fungi were isolated and preserved by the Key Laboratory of National Forestry and Grassland Administration on Control of Artiffcial Forest Diseases and Pests in South China (Changsha, China). The three fungi strains were inoculated into 150 ml sterilized Potato Dextrose Agar (PDA) solution in Erlenmeyer flasks and cultured at a constant temperature of 28˚C for 5 days to obtain the fermentation filtrate. Healthy callus tissues of *A. sinensis* were treated with 100 μl of W-1, W-15, YMY fermentation filtrate, and sterile PDA, with each treatment replicated three times. The YMY (CK2) filtrate served as the positive control, while the sterile PDA (CK) was the negative control.

### Measurement of SOD activity in callus of *A. sinensis*

The callus tissues of *A. sinensis* were treated with the fermentation filtrates of W-1, W-15, YMY, and sterile PDA, and samples were collected at different time points (0, 4, 8, 12, 24, 48, 72, and 168 h) post-treatment to measure the levels of SOD. Approximately 0.2 g of the callus tissue was added to 1 mL of extraction solution and homogenized in an ice bath. The homogenate was centrifuged at 12,000 rpm for 10 minutes at 4˚C, and the supernatant was collected and kept on ice for testing SOD activity was measured according to the kit instructions (Suzhou Grace Biotechnology Co., Ltd). Each experiment was conducted three times. Each group was replicated three times.

### Measurement of POD activity in callus of *A. sinensis*

The callus tissues of *A. sinensis* were treated with the fermentation filtrates of W-1, W-15, YMY, and sterile PDA, and samples were collected at different time points (0, 4, 8, 12, 24, 48, 72, and 168 h) post-treatment. to measure the levels of POD. Approximately 0.2 g of the callus

tissue was added to 1 mL of extraction solution and homogenized in an ice bath. The homogenate was centrifuged at 12,000 rpm for 10 minutes at 4°C, and the supernatant was collected and kept on ice for testing. POD activity was measured according to the kit instructions (Suzhou Grace Biotechnology Co., Ltd). Each experiment was conducted three times. Each group was replicated three times.

### Isolation of total RNA and qRT-PCR

The callus tissues of *A. sinensis* were treated with the fermentation filtrates of W-1, W-15, YMY, and sterile PDA, and samples were collected at different time points (0, 4, 8, 12, 24, 48 h) post-treatment. Total RNA was extracted using the RNAprep Pure Plant Plus Kit (Beijing Tiangen Biotech Co., Ltd). Using the Evo M-MLV Kit (Changsha Accurate Biotechnology Co., Ltd), 4 μl of total RNA was reverse transcribed into cDNA according to the manufacturer's instructions for qRT-PCR analysis. PCR amplification was performed on the QuantStudio 3 (Thermo Fisher Scientific, China) using SYBR Green Pro Tag HS (Changsha Accurate Biotechnology Co., Ltd). The PCR cycling conditions were the same as described by Yu et al. [22]. The glyceraldehyde-3-phosphate dehydrogenase (*GADPH*) gene was used as an internal control. The experiment included three independent biological replicates, and the relative expression levels of *HMGS, DXR*, and *ASS-1* were calculated using the $2^{-\Delta\Delta CT}$ method [23]. The primers used for qRT-PCR are listed in Table 1.

### Validation of the agarwood-inducing capability of two fungi strains

Seven-year-old *A. sinensis* (from the Chengmai State Forest Farm, Chengmai County, Hainan Province, located at 19°67′E, 110°01′N) were treated with 200 ml of fermentation filtrates from W-1, W-15, and YMY, with each treatment replicated three times, and the YMY fermentation filtrate served as the control(CK). Three months later, agarwood samples were obtained, and 3 grams of *A. sinensis* wood chips were weighed and extracted using an ultrasonic bath in an ice-water mixture. The analysis was conducted using a 7890A-5975C gas chromatograph (Agilent Technologies, USA) under the following conditions: initial temperature of 50°C held for 2 minutes, ramped to 170°C at 5°C/min, then increased to 310°C at 10°C/min and held for 10 minutes. Helium was used as the carrier gas at a flow rate of 1 mL/min, with an injection volume of 1 μL. The detected compounds were identified using the NIST11 mass spectrometry database and retention times.

### Statistical analyses

Statistical analyses were performed using IBM SPSS Statistics 25.0 software. Duncan's new multiple range test was used for significance analysis.

## Results

### Molecular biological identification of two strains

The ITS gene sequences amplified from strains W-1 and W-15 were obtained through sequencing and submitted to GenBank. The accession numbers for these sequences are

**Table 1. Primers used in qRT-PCR.**

| Forward primer | Sequence (5'-3') | Reverse primer | Sequence (5'-3') |
|---|---|---|---|
| *HMGS*-F | GTTGAAGTCCAGGCACGAGTTCC | *HMGS*-R | CCGTTGTCACAGGCAGTGTTCTC |
| *DXR*-F | CGAGCAGAACTGGTGGCAACA | *DXR*-R | TCGCTTACTGGGCATCCTTCAATC |
| *ASS-1*-F | AAGAAGATGAAGGAGATGATTGAGA | *ASS-1*-R | TTTCAATAGCATGACGCAACAAG |
| *GADPH*-F | CTGGTATGGCATTCCGTGTA | *GADPH*-R | AACCACATCCTCTTCGGTGTA |

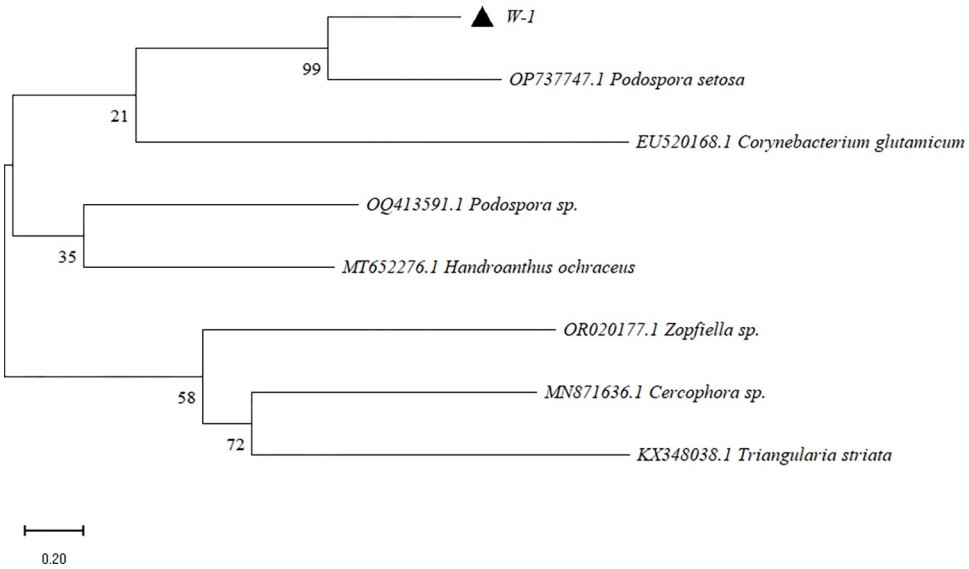

**Fig 1. W-1 phylogenetic tree.**

PP508237 and PP508252, respectively. Phylogenetic analysis showed that W-1 clustered with *Podospora setosa*, with a branch support rate of 99.00% (Fig 1). W-15 clustered with *Alternaria alstroemeriae*, with a branch support rate of 99.00% (Fig 2).

## Effect of two fungi on SOD activity in callus of *A. sinensis*

In this study, the SOD activity in callus treated with W-1 and W-15 showed an initial increase followed by a decrease, with the SOD activity in W-1 treatment peaking at 457.36 U/gmin$^{-1}$ at 48 h, and W-15 treatment peaking at 474.49 U/gmin$^{-1}$ at 8 h. This pattern of change is consistent with previous studies; in contrast, the SOD activity in YMY-treated callus initially decreased, but significantly increased after 24 h, reaching a peak of 388.89 U/gmin$^{-1}$ (Fig 3). This differing pattern of change may be related to the different active components or strains in

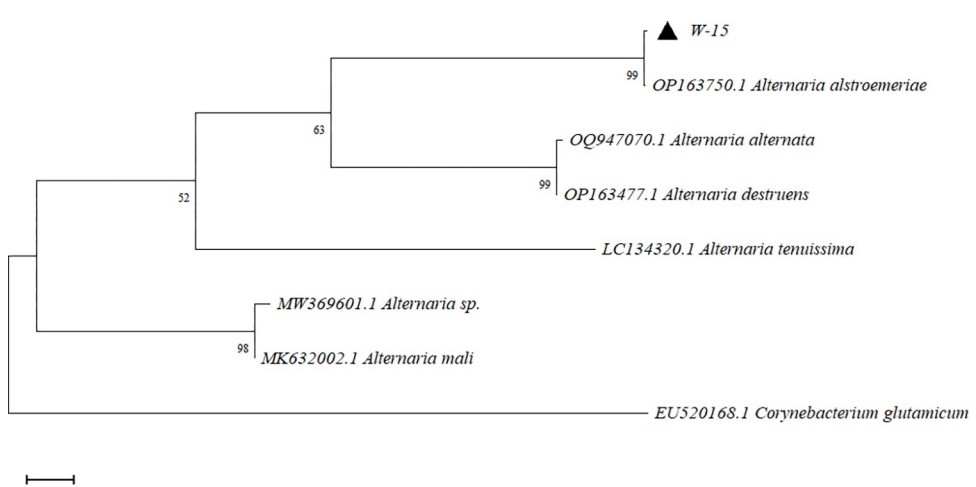

**Fig 2. W-15 phylogenetic tree.**

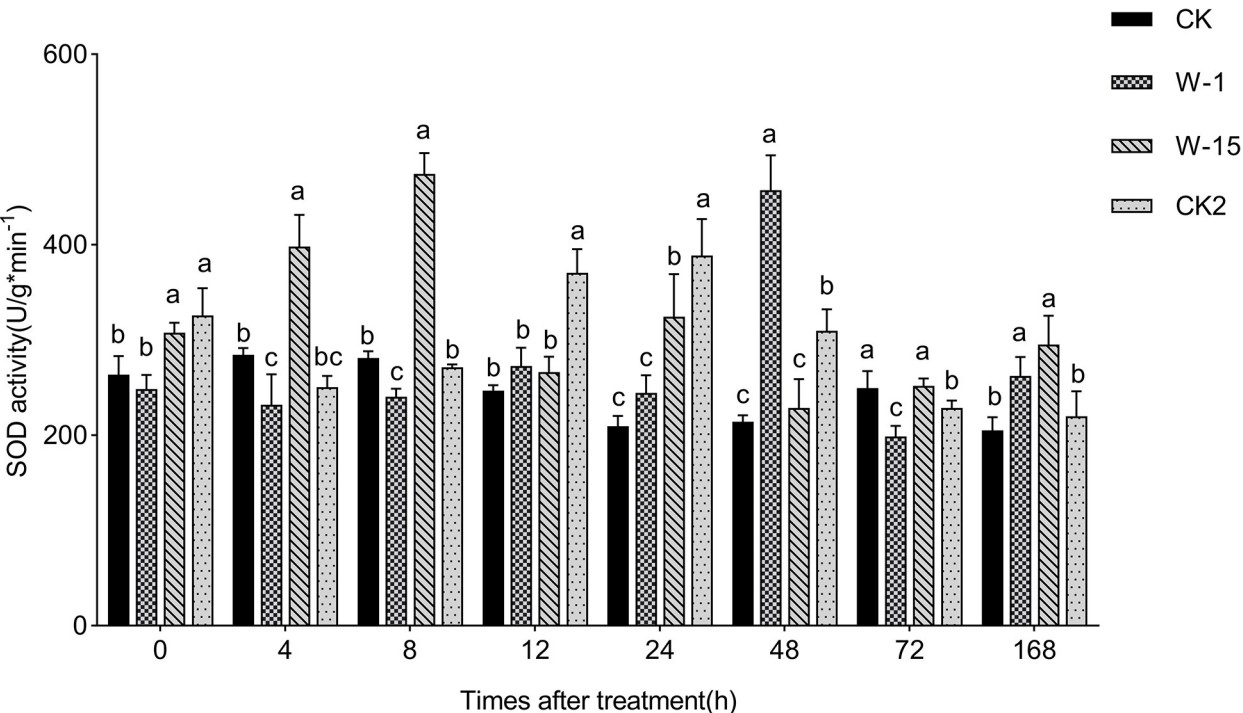

**Fig 3. Variations in SOD activity within callus.** Error bars represent standard deviation. Different lowercase letters indicate significant differences between treatments (P<0.05).

the YMY fermentation filtrate. It is worth noting that the SOD activity in the sterilized PDA treatment showed no significant change, further confirming that W-1, W-15, and YMY all can induce changes in SOD activity in *A. sinensis* callus.

## Effect of two fungi on POD activity in callus of *A. sinensis*

In this study, the POD activity in callus treated with W-1 peaked at 220.50 U/gmin$^{-1}$ at 12 h. In contrast, the POD activity in W-15 treated callus reached a maximum of 211.16 U/gmin$^{-1}$ at 48 h. Differently, the POD activity in YMY-treated callus peaked at 4 hours, then decreased, but rose again to the highest value of 230.83 U/gmin$^{-1}$ at 48 h (Fig 4). The results indicate that both fungi could significantly alter POD activity in *A. sinensis* callus.

## Effects of two fungi on the expression of *HMGS*, *DXR* and *ASS-1* genes in the callus of *A. sinensis*

The qRT-PCR results indicated that after treatment with W-1 and W-15 fermentation filtrates, the relative expression levels of the *HMGS*, *DXR*, and *ASS-1* genes were higher than those in the sterilized PDA treatment group and were similar to those observed in the YMY fermentation filtrate treatment group (Fig 5).

Under W-1 fermentation filtrate treatment, the expression level of the *HMGS* gene initially increased and then decreased over the course of the experiment, peaking at 4 h, reaching 3-fold that of the control group. Under W-15 fermentation filtrate treatment, the expression trend of the *HMGS* gene was similar to that under W-1 treatment, also peaking at 4 h, at 2.5-fold the level of the control group. Under YMY fermentation filtrate treatment, *HMGS* gene expression peaked at 48 h, reaching 3-fold that of the control group (Fig 5A) The

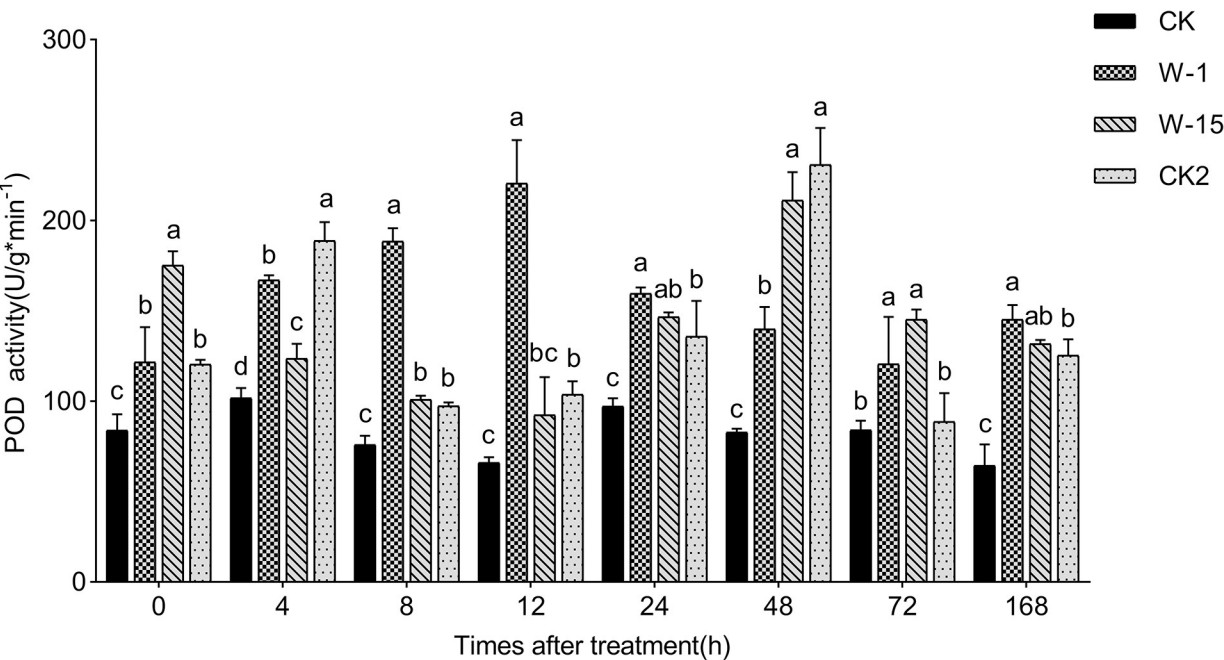

**Fig 4. Variations in POD activity within callus.** Error bars represent standard deviation. Different lowercase letters indicate significant differences between treatments (P<0.05).

expression levels of the *DXR* gene peaked at 0 h under W-1, W-15, and YMY fermentation filtrate treatments, reaching 5-fold, 7.5-fold, and 6.5-fold that of the control group, respectively, before gradually decreasing (Fig 5B).

Under W-1 fermentation filtrate treatment, *ASS-1* gene expression peaked at 8 h, reaching 120-fold that of the control group, and this trend continued. Under W-15 fermentation filtrate treatment, *ASS-1* gene expression showed a trend of rising, then falling, and rising again, peaking at 8 h at 130-fold that of the control group. Under YMY fermentation filtrate treatment, *ASS-1* gene expression peaked at 12 h, reaching 70-fold that of the control group (Fig 5C).

These results indicate that the metabolic pathways in *A. sinensis* callus were activated after treatment with W-1 and W-15, significantly enhancing the activity of related metabolic enzymes.

## Verification of the ability of two fungi to induce agarwood formation

To further verify whether *A. sinensis* callus can be used to screen for agarwood-inducing fungi, the samples were treated with W-1, W-15, and YMY fermentation filtrates for six months, resulting in a color change (Fig 6). GC-MS analysis revealed that 231, 305, and 242 compounds were detected in the samples treated with W-1, W-15, and YMY, respectively (S1–S3 Tables). The content of sesquiterpenes in the essential oils treated with W-1 and W-15 was 9.76% and 16.69%, respectively, while the content in the control group was 10.12% (Table 2). Four terpenoid compounds were common to all samples: Dehydrofukinone, Alloaromadendrene, Cycloheptane, 1-ethenyl-1-methyl-4-methylene-2-(2-methyl-1-propenyl)-, and Squalene. Among them, Dehydrofukinone and Alloaromadendrene exhibit antioxidant and antimicrobial activities [24–27], while Squalene is a triterpene with various applications in the food, chemical, and healthcare industries [28]. Aromatic compounds also contribute to the distinctive fragrance of agarwood. The content of aromatic compounds in the essential oils treated with W-1 and W-

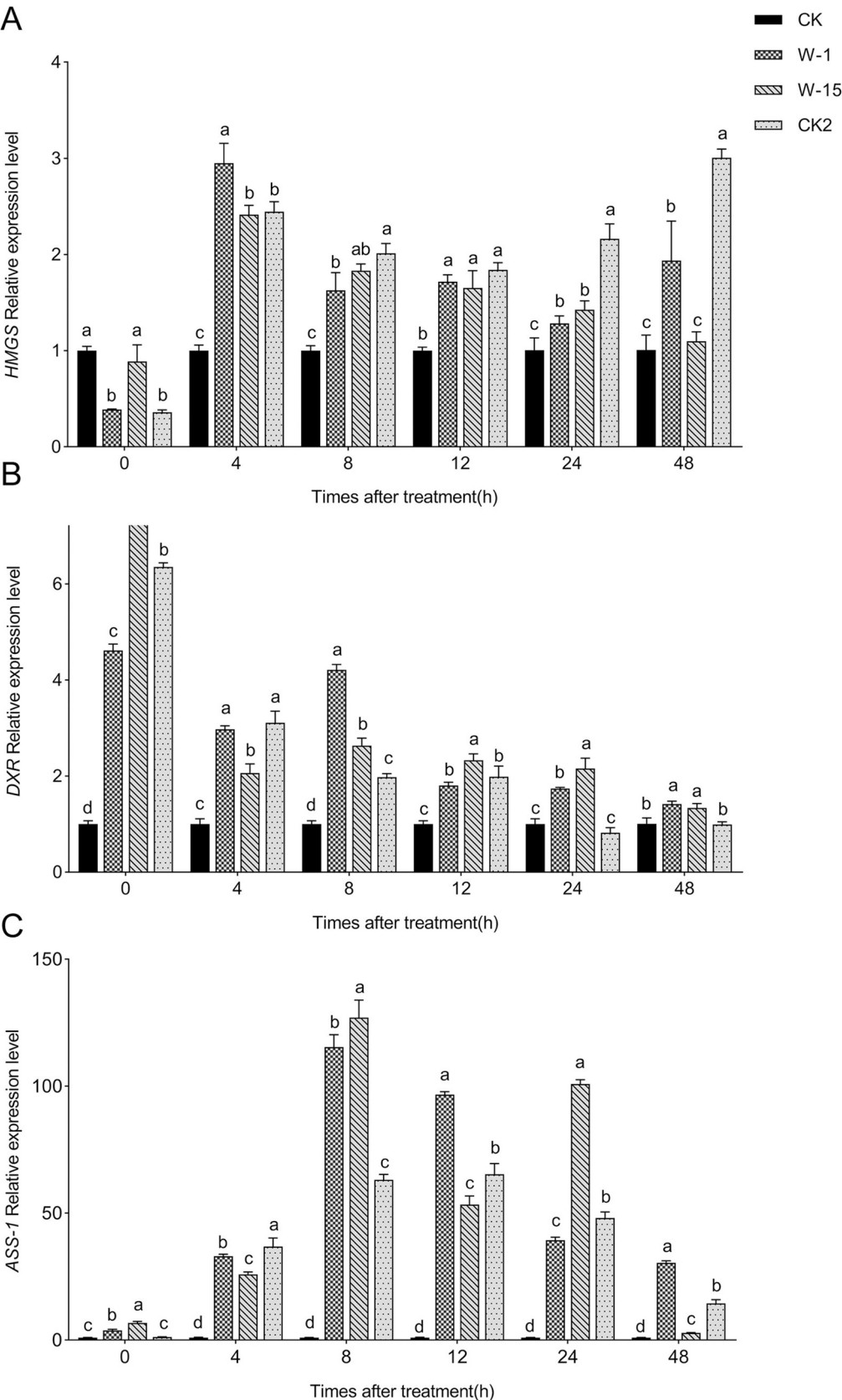

**Fig 5. Expression changes of genes in the callus.** (A) Changes of HMGS expression; (B) Changes of DXR expression; (C) Changes of ASS-1 expression. Error bars represent standard deviation. Different lowercase letters indicate significant differences between treatments (P<0.05).

15 was 21.61% and 21.6%, respectively, compared to 16.66% in the control group. Benzylacetone, which has antitussive, expectorant, and antiasthmatic effects [29], was detected in both the treatment and control groups. The content of chromone compounds was relatively low in the essential oils treated with W-1 and W-15, at 3.07% and 3.17%, respectively, compared to 4.38% in the control group. The experimental results indicate that W-1 and W-15 have the ability to induce agarwood formation.

## Discussion

Due to the slow and unpredictable process of agarwood formation in nature, recent research has increasingly focused on the artificial production of agarwood through fungi induction. Fungi serve as natural inductors, facilitate the accumulation of agarwood in a process that is both economically feasible and environmentally sustainable. However, only 8% of the fungi isolated from *Aquilaria* trees have been investigated for their capacity to enhance the production of agarwood [30,31]. Therefore, identifying fungi species capable of efficiently inducing agarwood formation is of great significance.

When plants are under stress, it leads to an explosion of ROS that disrupt the plant's internal balance [32]. Plants signal through hormones and ions to activate the downstream defense systems, such as antioxidant enzymes and secondary metabolite synthesis [33]. Finally, a large quantity of stored carbohydrates are synthesized into secondary metabolites [34], thereby assisting the plant in resisting stress. Zhu et al. used the endophytic fungi GG22 (*Fusarium oxysporum*) to treat *Rehmannia*, which increased the SOD and POD activities in *Rehmannia*, activated or enhanced the expression of related enzymes in primary metabolic pathways, and thereby increased the content of secondary metabolites [35]. Changes in defense enzyme activity are often accompanied by the accumulation of specific secondary metabolites. In the

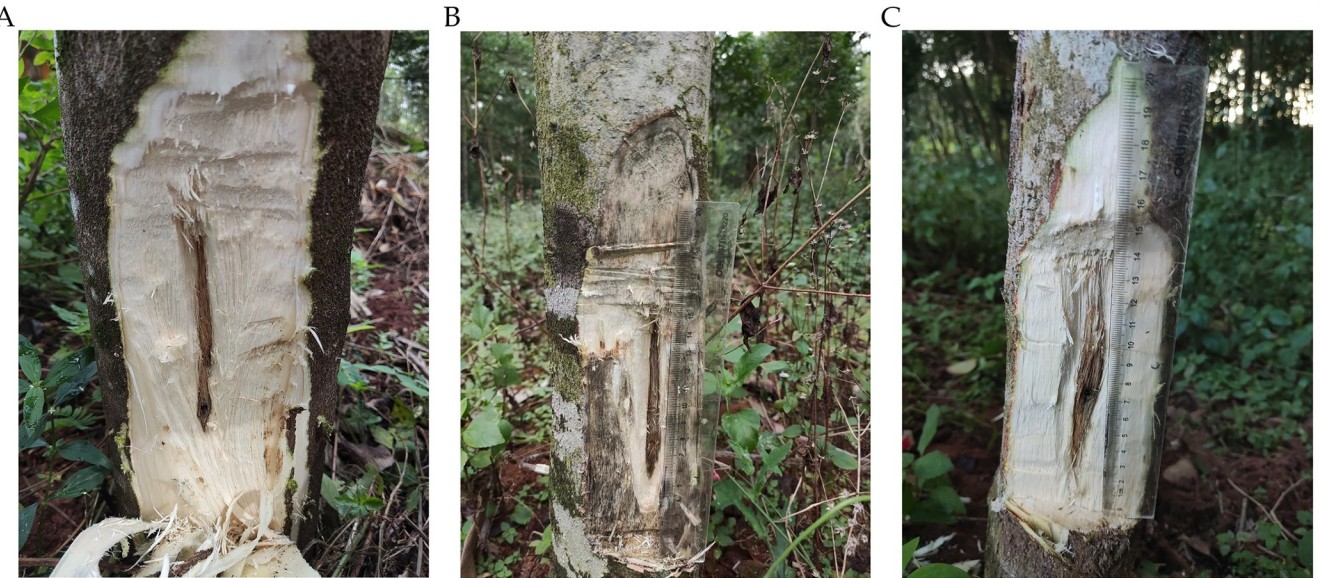

**Fig 6. Discoloration diagram of A. sinensis.** (A) W-1 treatment group; (B) W-15 treatment group; (C) YMY treatment group.

**Table 2. Terpenoids in different treatments.**

| No. | Compound | Relative amount % | | |
|---|---|---|---|---|
| | | W-1 | W-15 | YMY |
| 1 | alpha-Curcumene | 0.06 | 0.03 | - |
| 2 | 3,3,6,6,9,9-Hexamethyltetracyclo[6.1.0.02,4.05,7]nonane | 0.11 | - | - |
| 3 | beta-Patchoulene | 0.04 | - | - |
| 4 | (+)-valencene | 0.22 | - | 0.24 |
| 5 | (-)-Aristolene | 0.14 | - | - |
| 6 | alpha-Patchoulene | 0.13 | - | - |
| 7 | Diepicedrene-1-oxide | 0.04 | - | - |
| 8 | Aromadendrane | 0.06 | 0.27 | - |
| 9 | Solavetivone | 0.26 | - | - |
| 10 | Dehydrofukinone | 0.14 | 0.67 | 0.35 |
| 11 | beta-Selinene | 0.47 | 0.55 | - |
| 12 | Alloaromadendrene | 0.74 | 0.67 | 1.41 |
| 13 | Bicyclo[5.3.0]decane,2-methylene-5-(1-methylvinyl)-8-methyl- | 0.18 | - | - |
| 14 | 2-(4a,8-Dimethyl-1,2,3,4,4a,5,6,7-octahydro-naphthalen-2-yl)-prop-2-en-1-ol | 0.36 | - | - |
| 15 | Z-beta-Guaiene | 0.54 | - | 0.38 |
| 16 | Aromandendrene | 0.21 | - | 0.37 |
| 17 | (+)-Eremophilene | 0.27 | - | 0.06 |
| 18 | (-)-cis-beta-Elemene | 1.48 | - | - |
| 19 | Cycloheptane, 1-ethenyl-1-methyl-4-methylene-2-(2-methyl-1-propenyl)- | 1.54 | 0.36 | 1.89 |
| 20 | beta-Ionone | 0.23 | - | - |
| 21 | 5-Octen-2-one,6-methyl-8-(2,6,6-trimethyl-1-cyclohexen-1-yl)- | 0.11 | - | - |
| 22 | Longifolenaldehyde | 1.25 | 0.32 | - |
| 23 | Caryophyllene oxide | 0.14 | 0.27 | - |
| 24 | Squalene | 0.34 | 2.56 | 0.23 |
| 25 | Cholestan-3-ol,6-methyl-, (3.beta.,5.alpha.,6.alpha.)- | 0.37 | - | - |
| 26 | Neoisolongifolene, 8,9-dehydro- | - | 0.04 | 0.02 |
| 27 | 4-methyl-1-prop-1-en-2-ylcyclohexene | - | - | 0.26 |
| 28 | beta-Maaliene | - | 0.21 | 0.14 |
| 29 | (-)-alpha-Gurjunene | - | - | 0.01 |
| 30 | Hinesol | - | - | 0.08 |
| 31 | Longifolene | - | - | 0.17 |
| 32 | Bicyclo[4.4.0]dec-1-ene,2-isopropyl-5-methyl-9-methylene- | - | - | 0.29 |
| 33 | gamma-Gurjunene | - | - | 0.23 |
| 34 | (-)-Globulol | - | - | 0.14 |
| 35 | Cycloisolongifolene, 8,9-dehydro- | - | - | 0.08 |
| 36 | 1,8-dimethyl-4-(1-methylethyl)-spiro[4.5]dec-8-en-7-one | - | - | 0.28 |
| 37 | Culmorin | - | - | 0.15 |
| 38 | beta-EUDESMOL | - | - | 0.12 |
| 39 | Alpha-Farnesene | - | - | 0.11 |
| 40 | cis-Thujopsene | - | - | 0.35 |
| 41 | gamma-Elemene | - | - | 0.19 |
| 42 | γ-Gurjunepoxide-(2) | - | 2.48 | 0.85 |
| 43 | Guaia-1(10),11-diene | - | - | 0.32 |
| 44 | beta-Ionone | - | - | 0.34 |
| 45 | Procerin | - | - | 0.36 |
| 46 | Aromadendrene oxide 2 | - | 0.72 | 0.08 |

*(Continued)*

**Table 2.** (Continued)

| No. | Compound | Relative amount % | | |
|---|---|---|---|---|
| | | W-1 | W-15 | YMY |
| 47 | beta-Apo-13-carotenone | - | - | 0.06 |
| 48 | Limonene dioxide | - | - | 0.05 |
| 49 | Ajmaline | - | - | 0.08 |
| 50 | Methyl sandaracopimarate | - | 0.06 | 0.43 |
| 51 | Benzylideneacetone | - | 0.04 | - |
| 52 | Camphene | - | 0.01 | - |
| 53 | (+)-trans-Isolimonene | - | 0.16 | - |
| 54 | alpha.-Santalol | - | 0.68 | - |
| 55 | Eudesma-4(14),7(11)-diene | 0.16 | 0.06 | - |
| 56 | alpha-Elemene | - | 0.53 | - |
| 57 | alpha-Himachalene | - | 0.17 | - |
| 58 | 2-Methoxy-4-methyl-bicyclo[3.2.1]oct-2-ene | - | 0.23 | - |
| 59 | Aromadendrene, dehydro- | - | 0.14 | - |
| 60 | o-Mentha-1(7),8-dien-3-ol | 0.08 | 0.32 | - |
| 61 | Ledene oxide-(I) | - | 0.17 | - |
| 62 | Caryophyllene | - | 0.27 | - |
| 63 | Cedran-diol, 8S,13- | - | 0.25 | - |
| 64 | Caryophyllene-(I1) | - | 0.23 | - |
| 65 | gamma-Neoclovene | - | 3.15 | - |
| 66 | Isoaromadendrene epoxide | - | 0.61 | - |
| 67 | Farnesyl acetate | - | 0.23 | - |
| 68 | (2Z,4E)-3,7,11-Trimethyl-2,4,10-dodecatriene | - | 0.12 | - |
| 69 | viridiflorene | - | 0.12 | - |
| 70 | farnesol | - | 0.15 | - |
| 71 | 7-epi-cis-sesquisabinene hydrate | - | 0.11 | - |
| TOTAL | | 9.76 | 16.69 | 10.12 |

- not detected.

process of agarwood formation, fungi infection induces the activity of antioxidant and defense enzymes, promoting the synthesis of terpenoids and phenylpropanoids, which are key aromatic compounds in agarwood [36]. Similarly, in other plants, fungi infection can promote the accumulation of metabolites such as flavonoids, coumarins, and alkaloids by regulating the activity of defense enzymes [12,37]. In this study, the activities of SOD and POD were enhanced after treatment with W-1 and W-15. The results showed that W-1 and W-15 can induce the necessary defense responses by activating the activity of related enzymes.

In the process of agarwood formation, mechanical damage or other stressors activate specific genes and metabolic pathways, which ultimately result in the production of secondary metabolites like sesquiterpenes and 2-(2-phenylethyl) chromone [38]. The metabolic pathway of terpenoids has been extensively researched, but the biosynthesis of chromone derivatives has received relatively less attention [39]. Sesquiterpenes are synthesized by multiple enzymes via the MVA pathway in the cytoplasm and the MEP pathway in the plastids [40]. Liu et al. found that the fungi *Phaeoacremonium rubrigenum* increases sesquiterpene biosynthesis by upregulating enzymes in the MVA pathway of *A. sinensis*[41]. Zhang et al. showed that methyl jasmonate and hydrogen peroxide can increase the expression of the *SaDXR* gene in *Santalum album*., thereby enhancing the biosynthesis of sesquiterpenes [42]. In this study, treatment

with W-1 and W-15 significantly increased the expression levels of the *HMGS* and *DXR* genes in the callus tissues of *A. sinensis*. This upregulation might be part of the callus's defense response to fungi infection, thereby promoting agarwood formation [41]. The expression pattern of the *ASS-1* gene was consistent with previous findings, showing minimal expression in callus tissues treated with sterile PDA but significant expression following W-1 and W-15 treatments. The experimental results showed that the activities of POD, SOD, and the expression levels of the *ASS-1*, *HMGS*, and *DXR* genes in the experimental groups and the positive control group were all higher than those in the negative control group, indicating that W-1 and W-15 have similar capabilities to YMY in inducing agarwood production.

The effectiveness of these fungi in inducing agarwood formation was further confirmed by GC-MS analysis. The results showed a significant increase in the variety of volatile oil compounds in *A. sinensis* treated with W-1 and W-15, particularly in the content of sesquiterpenes and aromatic compounds. The sesquiterpene content in the W-1 treated samples was 9.76%, in W-15 treated samples 16.69%, compared to 10.12% in the control group. These results suggest that W-1 and W-15 can effectively induce agarwood formation. Moreover, the content of aromatic compounds in the W-1 and W-15 treated samples was significantly higher than in the control group, further confirming the effectiveness of these fungi in promoting agarwood formation. The results confirmed that fungi W-1 and W-15, similar to YMY, possess the ability to induce agarwood production. In conclusion, using callus tissue to assess the ability of fungi to induce agarwood formation is a rapid and reliable method. This approach not only enables the rapid determination of the ability of fungi to promote agarwood formation in the laboratory, but also contributes to the protection of wild *A. sinensis*.

## Conclusions

This study explored the feasibility of using callus tissue to evaluate the potential of fungi to induce agarwood formation. The experimental results showed that W-1 and W-15 treatments significantly enhanced the activity of SOD and POD in the callus tissues, while also upregulating the expression of key metabolic enzymes such as *HMGS*, *DXR*, and *ASS-1*. These changes suggest that W-1 and W-15 have similar potential to YMY in inducing agarwood formation. Moreover, GC-MS analysis revealed a significant increase in the content of sesquiterpenes and aromatic compounds in the samples treated with W-1 and W-15, further validating the effectiveness of these two fungi in inducing agarwood formation. Therefore, this study demonstrates that using callus tissue as a method for rapidly screening fungi that promote agarwood formation is feasible and effective. This method not only offers new insights for screening fungi resources that efficiently induce agarwood formation but also contributes to the protection and sustainable use of wild *A. sinensis* resources.

## Supporting information

**S1 Fig. Callus of *A. sinensis*.**
(TIF)

**S1 Table. GS-MS results of W-1 treatment.**
(PDF)

**S2 Table. GS-MS results of W-15 treatment.**
(PDF)

**S3 Table. GS-MS results of YMY treatment.**
(PDF)

## Author Contributions

**Conceptualization:** Guoying Zhou, Junang Liu.

**Formal analysis:** Yandong Xia, Zhuang Du.

**Methodology:** Zhikai Wang, Jungang Chen, Xinyu Miao.

**Project administration:** Guoying Zhou, Junang Liu.

**Writing – original draft:** Zhikai Wang.

**Writing – review & editing:** Zhikai Wang.

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
