## [Decision Letter · Decision Letter 0]

7 Jun 2024

PONE-D-24-13425Screening of agarwood-inducing fungi by the callus of Aquilaria sinensisPLOS ONE

Dear Dr. Liu,

Thank you for submitting your manuscript to PLOS ONE. There are some points highlighted by the independent reviewers which needs to be addressed before I can take any further decisions on your manuscript. Therefore, we invite you to submit a revised version of the manuscript that addresses the points raised during the review process.

We look forward to receiving your revised manuscript.

Kind regards,

Pankaj Bhardwaj, Ph.D.

Academic Editor

PLOS ONE

“This research was funded by National Key R & D Program of China (2023YFD1401304).”

“This research was funded by National Key R & D Program of China (2023YFD1401304).”

 “This research was funded by National Key R & D Program of China (2023YFD1401304).”

5. We note that your Data Availability Statement is currently as follows: [All relevant data are within the manuscript and its Supporting Information files.]

6. PLOS requires an ORCID iD for the corresponding author in Editorial Manager on papers submitted after December 6th, 2016. Please ensure that you have an ORCID iD and that it is validated in Editorial Manager. To do this, go to ‘Update my Information’ (in the upper left-hand corner of the main menu), and click on the Fetch/Validate link next to the ORCID field. This will take you to the ORCID site and allow you to create a new iD or authenticate a pre-existing iD in Editorial Manager. Please see the following video for instructions on linking an ORCID iD to your Editorial Manager account: https://www.youtube.com/watch?v=_xcclfuvtxQ.

Reviewers' comments:

Reviewer's Responses to Questions

**Comments to the Author**

1. Is the manuscript technically sound, and do the data support the conclusions?

Reviewer #1: Yes

Reviewer #2: Partly

Reviewer #3: Partly

2. Has the statistical analysis been performed appropriately and rigorously? 

Reviewer #1: Yes

Reviewer #2: N/A

Reviewer #3: No

3. Have the authors made all data underlying the findings in their manuscript fully available?

Reviewer #1: Yes

Reviewer #2: Yes

Reviewer #3: Yes

4. Is the manuscript presented in an intelligible fashion and written in standard English?

Reviewer #1: Yes

Reviewer #2: Yes

Reviewer #3: Yes

5. Review Comments to the Author

Reviewer #1: Aquilaria sinensis can produced agarwood induced by fungi, wound and so on. To find which species of fungi can high effective induce agarwood formation is important. In this manuscript, the authors revealed two fungi can changed the superoxide dismutase (SOD) and peroxidase (POD) activities as well as gene expression related to secondary metabolites using Quantitative Real-time PCR. However, there are some points need to be further considered.

1.The title can’t match the studies carried out in this manuscript.

2.The writing is very confused. In abstract, the experiments were not clearly described such as what is W-1 and W-15 (line 21). The writing of genes is wrong.

3.The introduction is too long and missed the main points. It needs to be focused on how fungi introduce the agarwood formation and why you carried out superoxide dismutase (SOD) and peroxidase (POD) activities. Why selected these genes to carried out by Quantitative Real-time PCR.

4.The study isolated fungi from A. sinensis in Chengmai County, Hainan Province, China. How and why to isolated them from Chengmai County?

5.These fungi were isolated and pre-served by the Key Laboratory of National Forestry and Grassland Administration on Control of Artiffcial Forest Diseases and Pests in South China (Changsha, China). What is the difference between the fungi from Chengmai County? Why did it differently?

6.Why set the experiments as W-1, W-15, and YMY (CK2), as well as sterile PDA (CK). What is the YMY (CK2), as well as sterile PDA (CK) and why?

7.What is the logic relationship between the callus of A. sinensis and the trees? Are the results of callus suitable for the tree?

8.The repeats of the experiments are unclear.

9.The results and conclusion need to be reorganized and make it sense.

10.The figures need to be reorganized to be more informative. Such as, only one figure in figure 1, why set it? What did the authors mean?

Lastly, to find which fungi can induce agarwood formation is meaningful, however the writings and organization in this manuscript need to further improved.

Reviewer #2: 1. Lacks Novelty

2. Need some more experimentation

3. The manuscript doesn't highlight what new knowledge has been acquired with the set of experiments performed

4. The fungi used in the study are not from the Plant-Microbe interaction microbiota, so explanation needed for using them at length, which is totally missing.

5. Only one core sesquiterpenoid gene expression has been studied, why?

6.The study is very basic in nature and needs some more value addition to meet the standards of the journal.

Reviewer #3: The data is not well correlated and statistically analyze. Discussion too surface and not well justified. Correlation of the fungi with chemical data is required for more scientifically discuss. Need more improvement in preparing the manuscript to be cited.

6. PLOS authors have the option to publish the peer review history of their article (what does this mean?). If published, this will include your full peer review and any attached files.

Reviewer #1: No

Reviewer #2: No

Reviewer #3: No

---

## [Author Response · Author response to Decision Letter 0]

26 Aug 2024

The point-by-point replies to the reviewers’ comments are listed below. 

Reviewers' Comments & Our Responses:

Reviewer #1: 

Aquilaria sinensis can produced agarwood induced by fungi, wound and so on. To find which species of fungi can high effective induce agarwood formation is important. In this manuscript, the authors revealed two fungi can changed the superoxide dismutase (SOD) and peroxidase (POD) activities as well as gene expression related to secondary metabolites using Quantitative Real-time PCR. However, there are some points need to be further considered.

Response: Special thanks to the reviewer for the positive evaluation.

1.The title can’t match the studies carried out in this manuscript. 

Response: Thank you for the comment. We changed the title as：“ Research on Using Aquilaria sinensis Callus to Evaluate the Agarwood-Inducing Potential of Fungi ”Thanks again for your helpful suggestions!

2.The writing is very confused. In abstract, the experiments were not clearly described such as what is W-1 and W-15 (line 21). The writing of genes is wrong. 

Response: Thank you for the comment. We acknowledge that the expression in the abstract may not have been clear, especially regarding the descriptions of W-1 and W-15. There were also errors in the gene nomenclature, for which we sincerely apologize. Therefore, we have rewritten the abstract to ensure that the experimental design and key points are clearly articulated. The gene names will be written in the correct standard format. 

The revised abstract is as follows: “Agarwood is a precious resinous heartwood highly valued for its cultural, religious, and medicinal significance. With the increasing market demand, natural agarwood resources are rapidly depleting, making the development of effective artificial induction methods for agarwood highly significant. This study aims to explore the feasibility of using callus tissue to assess the ability of fungi to induce agarwood formation. We selected two fungi isolated from Aquilaria sinensis, W-1 (Podospora setosa) and W-15 (Alternaria alstroemeriae), and used the known agarwood-inducing fungi YMY (Pestalotiopsis sp.) as a positive control, by treating A. sinensis callus with their fermented filtrates. The experimental results showed that W-1 and W-15 treatments significantly enhanced the activity of superoxide dismutase (SOD) and peroxidase (POD) in the callus tissue and upregulated the expression of 3-hydroxy-3-methylglutaryl-CoA synthase (HMGS), 1-deoxy-D-xylulose-5-phosphate reductoisomerase (DXR), and sesquiterpene synthase (ASS-1). The experimental results showed that W-1 and W-15 treatments significantly enhanced the activity of superoxide dismutase (SOD) and peroxidase (POD) in the callus tissue and upregulated the expression of 3-hydroxy-3-methylglutaryl-CoA synthase (HMGS), 1-deoxy-D-xylulose-5-phosphate reductoisomerase (DXR), and sesquiterpene synthase (ASS-1). In conclusion, this study demonstrates that using callus tissue to screen fungi capable of inducing agarwood is feasible and effective, providing new insights for optimizing agarwood production techniques in the future.”

3.The introduction is too long and missed the main points. It needs to be focused on how fungi introduce the agarwood formation and why you carried out superoxide dismutase (SOD) and peroxidase (POD) activities. Why selected these genes to carried out by Quantitative Real-time PCR. 

Response: Thank you for the comment. We have revised the introduction and provided an explanation for why we conducted the superoxide dismutase (SOD) and peroxidase (POD) activity studies, as well as the reasons for selecting these genes for Quantitative Real-time PCR analysis.

“Fungi, as the key factor in inducing agarwood formation, stimulate agarwood trees to produce defensive secondary metabolites through intricate physiological and biochemical mechanisms. In a natural environment, when agarwood trees are infected by fungi, the tree recognizes the fungi invasion and activates its defense response, a process considered the starting point of agarwood formation [9]. The core of the defense response lies in inducing stress responses through signaling pathways, leading to enhanced antioxidant enzyme activity, triggering the expression of a series of disease-resistant genes, and ultimately resulting in the accumulation of secondary metabolites [2]. These metabolites, including sesquiterpenes and 2-(2-phenylethyl) chromones, are the main active components of agarwood [10].

Fungi infection is often accompanied by oxidative stress responses in plant cells. Plant cells counter pathogen invasion by producing reactive oxygen species (ROS) such as superoxide anions (O₂⁻) [11]. To mitigate the toxicity of ROS, the antioxidant enzyme system within agarwood trees is rapidly activated. Superoxide dismutase (SOD), peroxidase (POD), and catalase (CAT) are the main antioxidant enzymes involved, working together to eliminate excess ROS and maintain redox balance within cells [12]. The enhancement of these enzyme activities is not only a direct response of plants to fungi invasion but also indirectly promotes the activation of secondary metabolic pathways related to agarwood formation [9]. Under fungi induction, the secondary metabolic pathways within agarwood trees are significantly activated, particularly the mevalonate (MVA) pathway and the methylerythritol phosphate (MEP) pathway. 3-Hydroxy-3-methylglutaryl-CoA synthase (HMGS) and 1-deoxy-D-xylulose-5-phosphate reductoisomerase (DXR) are the key rate-limiting enzymes in the MVA and MEP pathways, respectively [13, 14]. The upregulated expression of these enzymes promotes the increased synthesis of sesquiterpene compounds. ASS-1 is a typical wound-induced gene responsible for sesquiterpene formation in agarwood. Studies have found that ASS-1 expression is barely detectable in healthy A. sinensis callus or cell cultures, but its expression significantly increases in wounded tissues or jasmonic acid methyl ester-treated callus, leading to a corresponding increase in sesquiterpene compounds [15]. Fungi infection can significantly increase the accumulation of secondary metabolites in agarwood trees by regulating the gene expression of defense enzymes and these key enzymes, thereby enhancing the yield and quality of agarwood.”

4.The study isolated fungi from A. sinensis in Chengmai County, Hainan Province, China. How and why to isolated them from Chengmai County? 

Response: Thank you for the comment. The reason for isolating fungi from Chengmai County is that Chengmai is more suitable for the growth of Aquilaria sinensis compared to Changsha. Therefore, the fungi were isolated from Chengmai County. The method used for isolating the fungi was: 

“After culturing on Potato Dextrose Agar (PDA) plates at 25°C for 7 days, single fungi isolates were purified using hyphal tip isolation technique.”

5.These fungi were isolated and preserved by the Key Laboratory of National Forestry and Grassland Administration on Control of Artiffcial Forest Diseases and Pests in South China (Changsha, China). What is the difference between the fungi from Chengmai County? Why did it differently? 

Response: Thank you for the comment. There is no difference. Due to the lack of experimental conditions in Chengmai County, the fungi were isolated and preserved at the Key Laboratory of Forest Pest Control for Southern Plantations, National Forestry and Grassland Administration, in Changsha, China.

6.Why set the experiments as W-1, W-15, and YMY (CK2), as well as sterile PDA (CK). What is the YMY (CK2), as well as sterile PDA (CK) and why? 

Response: Thank you for the comment. W-1 and W-15 were used as experimental groups because their ability to induce agarwood formation was unknown. YMY, a fungi known for its ability to promote agarwood formation, served as the positive control, and sterile PDA was used to exclude any effects of the culture medium on the experiment. A more detailed description has also been provided in the manuscript. 

“In this study, we selected the fungi W-1 and W-15, whose ability to promote agarwood formation was not previously established, and used the known agarwood-inducing fungi YMY [19] as a positive control.”

7.What is the logic relationship between the callus of A. sinensis and the trees? Are the results of callus suitable for the tree? 

Response: Thank you for the comment. The logical relationship between the Aquilaria sinensis callus and the tree is described in the abstract. 

“The callus of A. sinensis is an undifferentiated cell mass induced from plant tissue through in vitro culture, and it shares physiological and biochemical similarities with the whole tree. The metabolic pathways and stress responses of callus tissue largely retain the characteristics of the tree, making it a common model for studying the formation mechanisms of agarwood [16, 17]. However, because callus tissue grows in a controlled environment, lacking the complex structure and environmental interactions found in mature trees, its research findings may require further validation and optimization when applied to actual trees [18].”

8.The repeats of the experiments are unclear. 

Response: Thank you for the comment. The number of replicates for each experiment, which was three, is detailed in the methods section of the manuscript.

9.The results and conclusion need to be reorganized and make it sense. 

Response: Thank you for the comment. We have reorganized the results and conclusions and also made revisions to the discussion section.

Effect of two fungi on SOD activity in callus of A. sinensis

“In this study, the SOD activity in callus treated with W-1 and W-15 showed an initial increase followed by a decrease, with the SOD activity in W-1 treatment peaking at 457.36 U/gmin-1 at 48 h, and W-15 treatment peaking at 474.49 U/gmin-1 at 8 h. This pattern of change is consistent with previous studies; in contrast, the SOD activity in YMY-treated callus initially decreased, but significantly increased after 24 h, reaching a peak of 388.89 U/gmin-1 (Fig 3). This differing pattern of change may be related to the different active components or strains in the YMY fermentation filtrate. It is worth noting that the SOD activity in the sterilized PDA treatment showed no significant change, further confirming that W-1, W-15, and YMY all can induce changes in SOD activity in A. sinensis callus.”

Effect of two fungi on POD activity in callus of A. sinensis

“In this study, the POD activity in callus treated with W-1 peaked at 220.50 U/gmin-1 at 12 h. In contrast, the POD activity in W-15 treated callus reached a maximum of 211.16 U/gmin-1 at 48 h. Differently, the POD activity in YMY-treated callus peaked at 4 h, then decreased, but rose again to the highest value of 230.83 U/gmin-1 at 48 h (Fig 4). The results indicate that both fungi could significantly alter POD activity in A. sinensis callus.”

Verification of the ability of two fungi to induce agarwood formation.

“To further verify whether Aquilaria sinensis callus can be used to screen for agarwood-inducing fungi, the samples were treated with W-1, W-15, and YMY (CK) fermentation filtrates for six months, resulting in a color change (Fig 4). GC-MS analysis revealed that 231, 305, and 242 compounds were detected in the samples treated with W-1, W-15, and YMY (CK), respectively (Supplementary Tables S1, S2, S3). The content of sesquiterpenes in the essential oils treated with W-1 and W-15 was 9.76% and 16.69%, respectively, while the content in the control group was 10.12% (Table 2). Four terpenoid compounds were common to all samples: Dehydrofukinone, Alloaromadendrene, Cycloheptane, 1-ethenyl-1-methyl-4-methylene-2-(2-methyl-1-propenyl)-, and Squalene.。Among them, Dehydrofukinone and Alloaromadendrene exhibit antioxidant and antimicrobial activities [23-26], while Squalene is a triterpene with various applications in the food, chemical, and healthcare industries [27]. Aromatic compounds also contribute to the distinctive fragrance of agarwood. The content of aromatic compounds in the essential oils treated with W-1 and W-15 was 21.61% and 21.6%, respectively, compared to 16.66% in the control group. Benzylacetone, which has antitussive, expectorant, and antiasthmatic effects [28], was detected in both the treatment and control groups. The content of chromone compounds was relatively low in the essential oils treated with W-1 and W-15, at 3.07% and 3.17%, respectively, compared to 4.38% in the control group. The experimental results indicate that W-1 and W-15 have the ability to induce agarwood formation.”

Conclusions

“This study explored the induction effects of W-1 and W-15 on agarwood formation by inoculating Aquilaria sinensis callus tissues with different fungi fermentation broths. The experimental results showed that W-1 and W-15 treatments significantly enhanced the activity of superoxide dismutase (SOD) and peroxidase (POD) in the callus tissues, while also upregulating the expression of key metabolic enzymes such as HMGS, DXR, and ASS-1. These changes suggest that W-1 and W-15 can promote the accumulation of agarwood-related compounds by activating antioxidant defense mechanisms and secondary metabolic pathways. Moreover, GC-MS analysis revealed a significant increase in the content of sesquiterpenes and aromatic compounds in the samples treated with W-1 and W-15, further validating the effectiveness of these two fungi in inducing agarwood formation. Therefore, this study demonstrates that using callus tissue as a method for rapidly screening fungi that promote agarwood formation is feasible and effective. This method not only offers new insights”

10.The figures need to be reorganized to be more informative. Such as, only one figure in figure 1, why set it? What did the authors mean?

Response: Thank you for the comment. Figure 1 shows the phylogenetic tree of W-1, and Figure 2 shows the phylogenetic tree of W-15. The titles will be highlighted when uploading the images. Thanks again for your helpful suggestions!

Finally, we would like to thank you once again for your valuable comments.

Reviewer #2: 

1. Lacks Novelty

Response: Thank you for the comment. We understand the reviewers' concern about the novelty of the study. During the revision process, we will emphasize the innovative aspects of our research, particularly the new insights reflected in the experimental design and result analysis. The introduction and discussion sections will be revised to highlight our study's unique contributions, including how examining SOD and POD activities and secondary metabolic gene expression reveals the potential mechanisms by which fungi influence agarwood formation.

2. Need some more experimentation

Response: Thank you for the comment. We understand the reviewers' suggestion to add more experiments. However, we regret that due to the limitation of experimental materials, it is not easy to add additional experiments. Nevertheless, we have conducted a more thorough analysis of the existing experiments and have improved the introduction and discussion sections, hoping to meet the standards for publication.

3. The manuscript doesn't highlight what new knowledge has been acquired with the set of experiments performed

Response: Thank you for the comment. In the revised manuscript, we will more clearly explain the new findings obtained through these experiments and their contribution to the existing literature. We have improved the abstract, introduction, discussion, and results sections to highlight the significance of this study. The following are the revisions made:

Abstract:

“Agarwood is a precious resinous heartwood highly valued for its cultural, religious, and medicinal significance. With the increasing market demand, natural agarwood resources are rapidly depleting, making the development of effective artificial induction methods for agarwood highly significant. This study aims to explore the feasibility of using callus tissue to assess the ability of fungi to induce agarwood formation. We selected two fungi isolated from Aquilaria sinensis, W-1 (Podospora setosa) and W-15 (Alternaria alstroemeriae), and used the known agarwood-inducing fungi YMY (Pestalotiopsis sp.) as a positive control, by

---

## [Decision Letter · Decision Letter 1]

17 Oct 2024

PONE-D-24-13425R1Research on Using Aquilaria sinensis Callus to Evaluate the Agarwood-Inducing Potential of FungiPLOS ONE

Dear Dr. Liu,

Thank you for submitting your manuscript to PLOS ONE. Please look in to the manuscript language, follow the highlighted shortfall by the reviewers and resubmit the manuscript. 

We look forward to receiving your revised manuscript.

Kind regards,

Pankaj Bhardwaj, Ph.D.

Academic Editor

PLOS ONE

Journal Requirements:

Reviewers' comments:

Reviewer's Responses to Questions

**Comments to the Author**

1. If the authors have adequately addressed your comments raised in a previous round of review and you feel that this manuscript is now acceptable for publication, you may indicate that here to bypass the “Comments to the Author” section, enter your conflict of interest statement in the “Confidential to Editor” section, and submit your "Accept" recommendation.

Reviewer #1: (No Response)

Reviewer #4: All comments have been addressed

Reviewer #5: All comments have been addressed

2. Is the manuscript technically sound, and do the data support the conclusions?

Reviewer #1: Yes

Reviewer #4: Yes

Reviewer #5: Yes

3. Has the statistical analysis been performed appropriately and rigorously? 

Reviewer #1: I Don't Know

Reviewer #4: Yes

Reviewer #5: Yes

4. Have the authors made all data underlying the findings in their manuscript fully available?

Reviewer #1: Yes

Reviewer #4: Yes

Reviewer #5: Yes

5. Is the manuscript presented in an intelligible fashion and written in standard English?

Reviewer #1: No

Reviewer #4: Yes

Reviewer #5: Yes

6. Review Comments to the Author

Reviewer #1: Although these authors responsed to my comments, the writing still need further impoved，such as genes need to be written in italics.

Reviewer #4: 1- Abbreviation of each word should be written once in the text of the introduction of the authors instead of the sod text and its full name.

2- “These changes suggest that W-1 and W-15 can promote the accumulation of agarwood-related compounds by activating antioxidant defense mechanisms and secondary metabolic pathways” On what basis do the authors make this claim?

Reviewer #5: Dear Editor!

I see that the authors have addressed reviewer comments in detail. I suggest its publicaiton after the followin minor improvements are made to the paper.

1. Clearly specifying the number of biological replicates and technical replicates used in each experiment can reinforce the reliability of the data.

2. The discussion could be strengthened by integrating more literature to explain the mechanisms by which the observed gene expression changes lead to agarwood formation. This might include discussing how increased superoxide dismutase (SOD) and peroxidase (POD) activities are linked to downstream processes in secondary metabolite production.

3. While the authors mention a few studies, a more thorough comparison with previous research findings would highlight the novelty and importance of their work. For instance, they could elaborate on how their findings align with or differ from past studies on fungal induction of agarwood.

4. While the authors have included information about the genes studied (e.g., HMGS, DXR, and ASS-1), explaining in greater detail why these specific genes were chosen (beyond their known functions) could add clarity. Additionally, discussing whether other genes related to secondary metabolite pathways were considered and why they were excluded might strengthen the rationale.

7. PLOS authors have the option to publish the peer review history of their article (what does this mean?). If published, this will include your full peer review and any attached files.

Reviewer #1: No

Reviewer #4: No

Reviewer #5: **Yes: **Tariq Khan

---

## [Author Response · Author response to Decision Letter 1]

14 Nov 2024

Dear editor,

Thank you very much for your kind letter and the reviewers’ comments concerning our manuscript entitled 'Research on Using Aquilaria sinensis Callus to Evaluate the Agarwood-Inducing Potential of Fungi' (ID: PONE-D-24-13425). The comments were very valuable and helpful for revising and improving our paper as well as for guiding our research. We have thoroughly considered the comments and substantially revised our manuscript. We have highlighted the revisions in the manuscript in red.

The point-by-point replies to the editor's comments are listed below.

Editor's Comments & Our Responses:

Response: Thank you for the comment. We have checked the references and ensured their completeness and accuracy! Additionally, we have added references to the discussion section according to the reviewer's suggestions and ensured their completeness and accuracy! Here are the additional references we added, located at lines 229, 232, 233, 234, 349, 351, and 252 of the article. Their respective numbers are 31, 32, 33, 34, 36, 38, 39, and 40.

“Fungi serve as natural inductors, facilitate the accumulation of agarwood in a process that is both economically feasible and environmentally sustainable. However, only 8% of the fungi isolated from Aquilaria trees have been investigated for their capacity to enhance the production of agarwood [30, 31].

When plants are under stress, it leads to an explosion of ROS that disrupt the plant's internal balance [32]. Plants signal through hormones and ions to activate the downstream defense systems, such as antioxidant enzymes and secondary metabolite synthesis [33]. Finally, a large quantity of stored carbohydrates are synthesized into secondary metabolites [34], thereby assisting the plant in resisting stress.

In the process of agarwood formation, mechanical damage or other stressors activate specific genes and metabolic pathways, which ultimately result in the production of secondary metabolites like sesquiterpenes and 2-(2-phenylethyl) chromone [38]. The metabolic pathway of terpenoids has been extensively researched, but the biosynthesis of chromone derivatives has received relatively less attention [39]. Sesquiterpenes are synthesized by multiple enzymes via the MVA pathway in the cytoplasm and the MEP pathway in the plastids [40].

31. Zhang X, Wang LX, Hao R, Huang JJ, Zargar M, Chen MX, et al. Sesquiterpenoids in Agarwood: Biosynthesis, Microbial Induction, and Pharmacological Activities. J Agric Food Chem. 2024;72(42):23039-52. Epub 2024/10/08. doi: 10.1021/acs.jafc.4c06383. PubMed PMID: 39378105.

32. Chen Q, Jin Y, Guo X, Xu M, Wei G, Lu X, et al. Metabolomic responses to the mechanical wounding of Catharanthus roseus' upper leaves. PeerJ. 2023;11:e14539. Epub 2023/03/28. doi: 10.7717/peerj.14539. PubMed PMID: 36968002; PubMed Central PMCID: PMCPMC10035419.

33. Ghorbel M, Brini F. Hormone mediated cell signaling in plants under changing environmental stress. Plant Gene. 2021;28:100335. doi: 10.1016/j.plgene.2021.100335.

34. Cui Z, Li X, Xu D, Yang Z. Changes in Non-Structural Carbohydrates, Wood Properties and Essential Oil During Chemically-Induced Heartwood Formation in Dalbergia odorifera. Front Plant Sci. 2020;11:1161. Epub 2020/09/10. doi: 10.3389/fpls.2020.01161. PubMed PMID: 32903589; PubMed Central PMCID: PMCPMC7438546.

36. Liu T, Liu Y, Fu Y, Qiao M, Wei P, Liu Z, et al. Structural, defense enzyme activity and chemical composition changes in the xylem of Aquilaria sinensis during fungus induction. Industrial Crops and Products. 2024;208:117804. doi: 10.1016/j.indcrop.2023.117804.

38. Zhang Z, Zhang X, Yang Y, Wei J-h, Meng H, Gao Z-h, et al. Hydrogen peroxide induces vessel occlusions and stimulates sesquiterpenes accumulation in stems of Aquilaria sinensis. Plant growth regulation. 2014;72:81-7. doi: 10.1007/s10725-013-9838-z.

39. Xu J, Du R, Wang Y, Chen J. Wound-Induced Temporal Reprogramming of Gene Expression during Agarwood Formation in Aquilaria sinensis. Plants (Basel). 2023;12(16). Epub 2023/08/26. doi: 10.3390/plants12162901. PubMed PMID: 37631113; PubMed Central PMCID: PMCPMC10459772.

40. Khodavirdipour A, Safaralizadeh R, Haghi M, Hosseinpourfeizi MA. Comparative de novo transcriptome analysis of flower and root of Oliveria decumbens Vent. to identify putative genes in terpenes biosynthesis pathway. Front Genet. 2022;13:916183. Epub 2022/08/23. doi: 10.3389/fgene.2022.916183. PubMed PMID: 35991569; PubMed Central PMCID: PMCPMC9386285.”

Finally, we would like to thank you once again for your valuable advice!

The point-by-point replies to the reviewers’ comments are listed below. 

Reviewers' Comments & Our Responses:

Reviewer #1: 

Although these authors responsed to my comments, the writing still need further impoved，such as genes need to be written in italics.

Response: Special thanks to the reviewer for the positive evaluation. We apologize for any errors in the details! We have rechecked the details of the article and modified the format of the genes. Thanks again for your helpful suggestions! Here are some of the changes made:

“The experimental results showed that W-1 and W-15 treatments significantly enhanced the activity of Superoxide dismutase (SOD) and Peroxidase (POD) in the callus tissue and upregulated the expression of 3-hydroxy-3-methylglutaryl-CoA synthase (HMGS), 1-deoxy-D-xylulose-5-phosphate reductoisomerase (DXR), and sesquiterpene synthase (ASS-1). The experimental results showed that W-1 and W-15 treatments significantly enhanced the activity of SOD and POD in the callus tissue and upregulated the expression of HMGS, DXR, and ASS-1.

The glyceraldehyde-3-phosphate dehydrogenase (GADPH) gene was used as an internal control. The experiment included three independent biological replicates, and the relative expression levels of HMGS, DXR, and ASS-1 were calculated using the 2-ΔΔCT method [22]. The primers used for qRT-PCR are listed in Table 1”

Finally, we would like to thank you once again for your valuable advice!

Reviewer #4: 

1. Abbreviation of each word should be written once in the text of the introduction of the authors instead of the sod text and its full name.

Response: Thank you for the comment. We re-reviewed the entire document and made necessary changes based on your feedback. Thank you again for your valuable suggestions！

2. “These changes suggest that W-1 and W-15 can promote the accumulation of agarwood-related compounds by activating antioxidant defense mechanisms and secondary metabolic pathways” On what basis do the authors make this claim?

Response: Thank you for the comment. I apologize for making an inaccurate statement. We have reviewed the entire article and made changes to the conclusion section in an effort to make our expression more accurate. The changes we made are in lines 281-282 of the article, and the specific changes are as follows:

“These changes suggest that W-1 and W-15 have similar potential to YMY in inducing agarwood formation.”

Once again, we sincerely apologize for our inaccurate description. Finally, we express our gratitude for your valuable suggestions！

Reviewer #5: 

Dear Editor!

I see that the authors have addressed reviewer comments in detail. I suggest its publicaiton after the followin minor improvements are made to the paper.

Response: Thank you for the positive feedback from the reviewers. We will make necessary revisions to our article based on their comments to meet the requirements for publication. Thanks again for your helpful suggestions!

1. Clearly specifying the number of biological replicates and technical replicates used in each experiment can reinforce the reliability of the data.

Response: Thank you for the comment. The number of replicates for each experiment, which was three, is detailed in the methods section of the manuscript. Thanks again for your helpful suggestions!

2. The discussion could be strengthened by integrating more literature to explain the mechanisms by which the observed gene expression changes lead to agarwood formation. This might include discussing how increased superoxide dismutase (SOD) and peroxidase (POD) activities are linked to downstream processes in secondary metabolite production.

Response: Thank you for the comment. Your input is highly valued by us. We have made modifications to the discussion section and included more references. The changes are located at lines 231-235 and 347-252 of the article, and here are the specific changes:

“When plants are under stress, it leads to an explosion of ROS that disrupt the plant's internal balance [32]. Plants signal through hormones and ions to activate the downstream defense systems, such as antioxidant enzymes and secondary metabolite synthesis [33]. Finally, a large quantity of stored carbohydrates are synthesized into secondary metabolites [34], thereby assisting the plant in resisting stress.

In the process of agarwood formation, mechanical damage or other stressors activate specific genes and metabolic pathways, which ultimately result in the production of secondary metabolites like sesquiterpenes and 2-(2-phenylethyl) chromone [38]. The metabolic pathway of terpenoids has been extensively researched, but the biosynthesis of chromone derivatives has received relatively less attention [39]. Sesquiterpenes are synthesized by multiple enzymes via the MVA pathway in the cytoplasm and the MEP pathway in the plastids [40].”

Finally, we would like to thank you once again for your valuable advice!

3. While the authors mention a few studies, a more thorough comparison with previous research findings would highlight the novelty and importance of their work. For instance, they could elaborate on how their findings align with or differ from past studies on fungal induction of agarwood.

Response: Thank you for the comment. Your input is highly valued by us. We emphasized in our discussion the scarcity of fungal resources with the ability to induce agarwood production, and also highlighted that using callus tissue to judge the fungal ability to promote agarwood formation is not only faster and more convenient, but also helps to protect wild Aquilaria sinensis. Our modifications are located at lines 226-229 and 275-276 of the article, and here are the specific changes we made:

“Fungi serve as natural inductors, facilitate the accumulation of agarwood in a process that is both economically feasible and environmentally sustainable. However, only 8% of the fungi isolated from Aquilaria trees have been investigated for their capacity to enhance the production of agarwood [30, 31].

This approach not only enables the rapid determination of the ability of fungi to promote agarwood formation in the laboratory, but also contributes to the protection of wild A. sinensis.”

Finally, we would like to thank you once again for your valuable advice!

4. While the authors have included information about the genes studied (e.g., HMGS, DXR, and ASS-1), explaining in greater detail why these specific genes were chosen (beyond their known functions) could add clarity. Additionally, discussing whether other genes related to secondary metabolite pathways were considered and why they were excluded might strengthen the rationale.

Response: Thank you for the comment. Your opinion has been very inspiring to us. The production of agarwood is not controlled by a single metabolic pathway or a single gene expression. It is the result of the combined action of multiple factors, which is why agarwood is so valuable. We explain the reasons for choosing HMGS, DXR, and ASS-1 in the abstract and discussion sections, respectively, located at 61-70 lines and 247-252 lines of the article. Here are the specific details:

“Under fungi induction, the secondary metabolic pathways within agarwood trees are significantly activated, particularly the mevalonate (MVA) pathway and the methylerythritol phosphate (MEP) pathway. 3-Hydroxy-3-methylglutaryl-CoA synthase (HMGS) and 1-deoxy-D-xylulose-5-phosphate reductoisomerase (DXR) are the key rate-limiting enzymes in the MVA and MEP pathways, respectively [13, 14]. The upregulated expression of these enzymes promotes the increased synthesis of sesquiterpene compounds. Sesquiterpene synthase (ASS-1) is a typical wound-induced gene responsible for sesquiterpene formation in agarwood. Studies have found that ASS-1 expression is barely detectable in healthy A. sinensis callus or cell cultures, but its expression significantly increases in wounded tissues or jasmonic acid methyl ester-treated callus, leading to a corresponding increase in sesquiterpene compounds [15].

In the process of agarwood formation, mechanical damage or other stressors activate specific genes and metabolic pathways, which ultimately result in the production of secondary metabolites like sesquiterpenes and 2-(2-phenylethyl) chromone [38]. The metabolic pathway of terpenoids has been extensively researched, but the biosynthesis of chromone derivatives has received relatively less attention [39]. Sesquiterpenes are synthesized by multiple enzymes via the MVA pathway in the cytoplasm and the MEP pathway in the plastids [40].”

Finally, we would like to thank you once again for your valuable advice!

We hope, with these modifications, the quality of our manuscript would meet the publication standard of the Journal. We appreciate very much for your time in editing our manuscript, and the detailed and useful comments and suggestions from you and reviewers. I am looking forward to hearing from your suggestion about our revised manuscript.

Sincerely,

Zhikai Wang

Central South University of Forestry and Technology

College of Life Sciences and Technology

No. 498, Shaoshan South Road, Tianxin District, Changsha City, Hunan Province

wwzk621@163.com

+86-18339259117

---

## [Decision Letter · Decision Letter 2]

9 Dec 2024

Research on Using Aquilaria sinensis Callus to Evaluate the Agarwood-Inducing Potential of Fungi

PONE-D-24-13425R2

Dear Dr. Liu,

We’re pleased to inform you that your manuscript has been judged scientifically suitable for publication and will be formally accepted for publication once it meets all outstanding technical requirements.

Kind regards,

Pankaj Bhardwaj, Ph.D.

Academic Editor

PLOS ONE

Additional Editor Comments (optional):

Reviewers' comments:

Reviewer's Responses to Questions

**Comments to the Author**

1. If the authors have adequately addressed your comments raised in a previous round of review and you feel that this manuscript is now acceptable for publication, you may indicate that here to bypass the “Comments to the Author” section, enter your conflict of interest statement in the “Confidential to Editor” section, and submit your "Accept" recommendation.

Reviewer #1: All comments have been addressed

Reviewer #4: All comments have been addressed

2. Is the manuscript technically sound, and do the data support the conclusions?

Reviewer #1: Yes

Reviewer #4: Yes

3. Has the statistical analysis been performed appropriately and rigorously? 

Reviewer #1: Yes

Reviewer #4: Yes

4. Have the authors made all data underlying the findings in their manuscript fully available?

Reviewer #1: Yes

Reviewer #4: Yes

5. Is the manuscript presented in an intelligible fashion and written in standard English?

Reviewer #1: Yes

Reviewer #4: Yes

6. Review Comments to the Author

Reviewer #1: (No Response)

Reviewer #4: It can be seen that the authors have addressed reviewer comments in detail. I suggest its publication

7. PLOS authors have the option to publish the peer review history of their article (what does this mean?). If published, this will include your full peer review and any attached files.

Reviewer #1: No

Reviewer #4: No

---

## [Editor Report · Acceptance letter]

15 Dec 2024

PONE-D-24-13425R2 

PLOS ONE

Dear Dr. Liu, 

I'm pleased to inform you that your manuscript has been deemed suitable for publication in PLOS ONE. Congratulations! Your manuscript is now being handed over to our production team.

Kind regards, 

on behalf of

Dr. Pankaj Bhardwaj 

Academic Editor

PLOS ONE